# Immunohistochemical Detection of PIEZO Ion Channels in the Human Carotid Sinus and Carotid Body

**DOI:** 10.3390/biom15030386

**Published:** 2025-03-07

**Authors:** Elda Alba, Yolanda García-Mesa, Ramón Cobo, Patricia Cuendias, José Martín-Cruces, Iván Suazo, Graciela Martínez-Barbero, José A. Vega, Olivia García-Suárez, Teresa Cobo

**Affiliations:** 1Instituto de Neurociencias Vithas, 28010 Madrid, Spain; eldamaselda@gmail.com; 2Servicio de Neurología, Hospital Clínico San Carlos, 28040 Madrid, Spain; 3Grupo SINPOS, Departamento de Morfología y Biología Celular, Universidad de Oviedo, 33003 Oviedo, Spain; garciamyolanda@uniovi.es (Y.G.-M.); cuendiaspatricia@uniovi.es (P.C.); pepe3214@gmail.com (J.M.-C.); ivan.suazo@uautonoma.cl (I.S.); gracielamartinezbarbero@gmail.com (G.M.-B.); garciaolivia@uniovi.es (O.G.-S.); 4Instituto de Investigación Sanitaria del Principado de Asturias (ISPA), 33011 Oviedo, Spain; 5Servicio de Otorrinolaringología, Hospital Universitario Marqués de Valdecilla, 39008 Santander, Spain; ramoncobodiaz@gmail.com; 6Facultad de Ciencias de la Salud, Universidad Autónoma de Chile, Santiago 8330015, Chile; 7Departamento de Cirugía y Especialidades Médico-Quirúrgicas, Universidad de Oviedo, 33003 Oviedo, Spain; teresacobo@uniovi.es; 8Instituto Asturiano de Odontología, 33006 Oviedo, Spain

**Keywords:** carotid sinus, carotid body, petrosal ganglion, superior cervical sympathetic ganglion, nerves, PIEZO ion channels, immunohistochemistry, human

## Abstract

The carotid sinus and the carotid body are major peripheral chemo- and baro(mechano)receptors that sense changes in arterial wall pressure and in oxygen, carbon dioxide, and pH in arterial blood. Recently, it was demonstrated that the PIEZO1 and PIEZO2 mechanoreceptor/mechanotransducers are responsible for the baroreflex in the murine aortic arch (aortic sinus). Furthermore, some experimental evidence suggests that the carotid body could participate in mechanosensing. In this study, we used immunohistochemistry and immunofluorescence in conjunction with laser confocal microscopy to study the distribution of PIEZO1 and PIEZO2 in the human carotid sinus and carotid body as well as in the petrosal ganglion of the glossopharyngeal nerve and the superior cervical sympathetic ganglion. PIEZO1 and PIEZO2 were detected in different morphotypes of sensory nerve formations in the walls of the carotid sinus and carotid artery walls. In the carotid body, PIEZO1 was present in a small population of type I glomus cells and absent in nerves, whereas PIEZO2 was present in both clusters of type I glomus cells and nerves. The most prominent expression of PIEZO1 and PIEZO2 in the carotid body was found in type II glomus cells. On the other hand, in the petrosal ganglion, around 25% of neurons were PIEZO1-positive, and around 85% were PIEZO2-positive; regarding the superior cervical sympathetic ganglion, around 71% and 86% displayed PIEZO1 and PIEZO2, respectively. The results of this study suggest that PIEZO1 and PIEZO2 could be involved in the detection and/or mechanotransduction of the human carotid sinus, whereas the role of the carotid body is more doubtful since PIEZO1 and PIEZO2 were only detected in some nerves and PIEZO2 was present in a small population of type I glomus cells, with PIEZO1 being absent in these cells. However, since immunoreactivity for PIEZO2 was detected in type II glomus cells, researchers should investigate whether these cells play a role in the detection of mechanical stimuli and/or participate in mechanotransduction.

## 1. Introduction

Most body cells are subject to the action of different forces and possess molecular mechanisms that allow them to sense these forces. The proteins that can be modified by the action of forces are collectively known as mechanosensors and include transmembrane proteins (especially ion channels [1]), intracellular proteins (especially cytoskeletal proteins [2]), and components of the extracellular matrix [3]. Among the transmembrane proteins involved in mechanotransduction, the most important are the PIEZO channels PIEZO1 and PIEZO2 [4,5,6]. Mechanosensors are organized into molecular complexes responsible for both the mechanosensing and mechanotransduction pathways. As such, mechanotransduction can be defined as the process whereby mechanical stimuli (e.g., compression, stiffness, elasticity, membrane tension, hydrostatic pressure, and shear stress) become biochemical and biological processes, and it is critical in numerous biological processes [7,8].

The carotid sinus and body are two structures localized at the bifurcation of the common carotid artery, and they are regarded as the primary baroreceptor and chemoreceptor, respectively. They detect changes in pressure on the artery walls and in arterial blood oxygen (hypoxemia), carbon dioxide (hypercapnia), and pH (acidosis) levels [9,10,11,12]. They consistently play key roles in regulating breathing and cardiovascular function [13].

The carotid sinus functions as a baroreceptor [14], although it has recently been proposed that the carotid sinus acts as a mechanotransducer of wall shear stress oscillation and not as a baroreceptor [15]. In any case, the molecular identities of these baroreceptors are still only partially known, but different ion channels have been identified as putative mechanoreceptors in the aortic and carotid sinuses. They include γ-epithelial sodium channels [16], acid-sensing ion channel 2 [17], transient receptor potential channel canonical 5 [18,19], tentonin [20], and the PIEZO channels [21,22]. Interestingly, the best-characterized mechanotransducers, PIEZO1 and PIEZO2, are present in the somas and terminals of the baroreceptors of the petrosal ganglion, and, acting together, they regulate the baroreflex in the carotid and aortic sinuses [21,22,23]. However, some scholars have questioned the role of PIEZO channels in baroreceptor function [24].

Regarding the carotid body, although it is primarily a chemosensor, there is evidence that it could also have a mechanosensory component, or at least that there is interdependence between the baroreceptor and chemoreceptor reflex responses [25,26]. Furthermore, neurons in the petrosal ganglion that innervate glomus cells express ion channels related to mechanosensitivity, including acid-sensing ion channel 2 [12] and PIEZO channels [21,22]. Nevertheless, as far as we know, the occurrence of PIEZO channels in the glomus cells or in the nerves supplying the carotid body has never been investigated.

Thus, the present study was designed, in which we investigated the presence of PIEZO ion channels in the human carotid sinus and carotid body of healthy participants. We used immunohistochemistry to study the distribution of PIEZO1 and PIEZO2 in the carotid sinus and carotid body, as well as in the petrosal ganglion and the sympathetic superior ganglion, which represent the localization of the afferences and efferences of both structures.

## 2. Materials and Methods

Tissue samples containing the common artery bifurcation, the carotid sinus, and the carotid body were obtained during removal of organs for transplantation from subjects who died in traffic accidents (Hospital Universitario Central de Asturias, Oviedo, Spain). There were 8 subjects (5 males and 3 females), with ages ranging between 38 and 68 years. The petrosal ganglia (*n* = 4) and the sympathetic superior ganglion (*n* = 4) were dissected and included in this study. The pieces were cleaned in 4 °C saline solution, fixed in 10% formaldehyde in 0.1 M phosphate-buffered saline (pH 7.4) for 24 h at 4 °C, dehydrated, and routinely embedded in paraffin.

These materials were obtained in compliance with the Spanish Law and the guidelines of the Declaration of Helsinki II. Sections of the same material were used in a previous study of the carotid body [12]. This material, which was used for research purposes, was deposited in the Department of Morphology and Cell Biology of the University of Oviedo, National Registry of Biobanks (Collections Section, Ref. C-0001627), created and authorized by the Ministry of Economy and Competitiveness of the Government of Spain on 30 November 2012.

### 2.1. Immunohistochemistry

Indirect peroxidase–anti-peroxidase immunohistochemistry was conducted as follows: sections of paraffin-embedded tissues (10 μm thick) were mounted on gelatin-coated microscope slides, and then they were rehydrated and rinsed in 0.05 M HCl Tris buffer (pH 7.5) containing 0.1% bovine serum albumin and 0.1% Triton X-100. Thereafter, the endogenous peroxidase activity (3% H_2_O_2_) and non-specific binding (10% fetal calf serum; F2442, Sima Aldrich, Saint Louis, MO, USA) were blocked, and the sections were incubated overnight in a humid chamber at 4 °C with primary antibodies against PIEZO1 and PIEZO2 (Table 1) diluted in a solution of Tris-HCl buffer (0.05 M, pH 7.5; Dako, Glostrup, Denmark) containing 0.1% bovine serum albumin, 0.2% fetal calf serum, and 0.1% Triton X-100. After incubation with the primary antibodies, the sections were rinsed in the same buffer as above and incubated with Dako EnVision System-labeled polymer-horseradish peroxidase anti-rabbit IgG or anti-mouse IgG (DakoCytomation, Dako) for 30 min at room temperature. Finally, the sections were washed, and their immunoreaction was visualized using 3-3′-diaminobenzidine as a chromogen. For control purposes, representative sections were processed as above using non-immune rabbit or mouse sera instead of the primary antibodies or by omitting the primary antibodies in the incubation. To visualize structural details, sections were slightly counterstained with hematoxylin and eosin.

The antibody against PIEZO1 was a polyclonal antibody developed against a synthetic peptide made of an internal portion of the human PIEZO1 protein between residues 1300 and 1350. The anti-PIEZO2 antibody was a polyclonal antibody developed against a synthetic peptide of human PIEZO2 with the following sequence: VFGFWAFGKHSAAADITSSLSEDQVPGPFLVMVLIQFGT MVVD RALYLRK.

### 2.2. Double Immunofluorescence

Sections were also processed for the simultaneous detection of PIEZO ion channels with neuronal markers (neurofilament protein (NFP) or neuron-specific enolase (NSE)), type I glomus cell markers (synaptophysin (SYN)), and support type II glial cell and Schwann cell (S100 protein (S100P)) markers [27,28]; type I glomus cells also display NSE [29]. Non-specific binding was reduced via incubation with a solution of 25% calf bovine serum in Tris buffer solution (TBS) for 30 min. The sections were incubated with a 1:1 *v*/*v* mixture of polyclonal antibodies against PIEZOs and monoclonal antibodies against NFP, NSE, S100P, or SYN in a humid chamber overnight at 4 °C. After being rinsed with TBS, the sections were incubated for one hour with CFL488-conjugated bovine anti-rabbit IgG (diluted 1:200 in TBS; sc-362260, Santa Cruz Biotechnology, Heidelberg, Germany) and then rinsed and incubated again for another hour with CyTM3-conjugated donkey anti-mouse antibodies (diluted 1:100 in TBS; Jackson-ImmunoResearch, Baltimore, MD, USA). Both steps were performed in a dark, humid chamber at room temperature, and washing with phosphate-buffered saline and Tween 20 was performed between both incubations. Finally, the immuno-stained samples were mounted with 4′,6-diamino-2-phenylindole (10 ng/mL) diluted in a Fluoromount-G mounting medium (Southern-Biotech, Birmingham, AL, USA) to contrast the nuclei. Specific reaction controls were created using the same method employed for single immunohistochemistry. Immunofluorescence was detected using a Leica TCS SP8 X confocal microscope along with a Leica DMI8 fluorescence microscope. Images were captured using the Leica Application Suite X acquisition program (version 1.8.1 Copyright 1997–2015 Leica Microsystems CMS GmbH, Wetzlar, Germany) at the Optical Microscopy and Image-Processing Unit of the University of Oviedo. The captured images were processed using Image J (version 1.43 g; Master Biophotonics Facility, Mac Master University Ontario; www.macbiophotonics.ca, accessed on 16 February 2025).

For control purposes, representative sections were processed in the same way as described above, but either by not using immune rabbit or mouse sera instead of primary antibodies or by omitting primary antibodies during incubation. Furthermore, when available, additional controls were developed using specifically preabsorbed antisera. Under these conditions, no positive immunostaining was observed.

### 2.3. Quantitative Analysis

A quantitative image analysis was carried out in the carotid sinus, carotid body, petrosal ganglia, and superior cervical sympathetic ganglia, which were processed for the immunodetection of PIEZOs using an automatic image analysis system (Quantimet 550, Leica, QWIN Program).

**Carotid sinus**. Measurements were made on 5 randomly selected fields per section (2.5 mm^2^), with five sections for the carotid sinus, spaced 200 µm apart (total 200 fields), that included the full thickness of the wall of the carotid sinus. The density of nerve profiles and sensory nerve formations with respect to PIEZO1 and PIEZO2 innervating the carotid sinus were expressed as the area occupied by the immunoreactivity.

**Carotid body**. Measurements were made in 10 randomly selected fields per section (5 mm^2^), with five sections for glomus spaced 100 µm apart (total 400 fields). The immunoreactive area for SYN was considered 100% of the area of type I cells; the areas occupied by the combination of SYN + PIEZO1 and SYN + PIEZO2 immunoreactivities were considered the area of type I cells with immunoreaction for PIEZO1 and PIEZO2, respectively. The results were expressed as the values of the means ± standard errors. Also, in 10 randomly selected fields per section (5 mm^2^), for five sections of the glomus that were spaced 100 µm apart (total 400 fields), the density of nerve profiles innervating the carotid body was determined based on the detection of NFP immunofluorescence; the area of immunoreactive NFP-positive nerve profiles was considered 100% of the area occupied by the nerve profiles; the area occupied by the combination of NFP + PIEZO1 and NFP + PIEZO2 was considered the area corresponding to nerves exhibiting immunoreaction for PIEZO1 and PIEZO2, respectively. The results were expressed as the area occupied by nerve profiles/mm^2^. Finally, in 10 randomly selected fields per section (5 mm^2^), 5 sections of the glomus spaced 100 µm apart (total 400 fields) and type I glomus cells and nerves were measured together via SES immunofluorescence. The immunoreactive area for NSE was considered 100% of the area of type I cells and nerves; the area occupied by the combination of NSE + PIEZO1 and NSE + PIEZO2 was considered the area of glomus type I cells plus nerves exhibiting immunoreaction for PIEZO1 and PIEZO2, respectively.

**Petrosal and superior cervical sympathetic ganglia.** The percentages of immunoreactive neurons for PIEZO1 and PIEZO2 were calculated in the petrosal ganglion of the glossopharyngeal nerve and superior cervical sympathetic ganglion. Counts were made in 5 randomly selected fields per section (2.5 mm^2^), with five sections for ganglia spaced 200 µm apart (the total was 200 fields: 100 fields for the petrosal ganglia and 100 fields for the sympathetic ganglia). The results were expressed as the percentage of immunoreactive neurons, considering 100% the number of neurons based on morphological criteria. No double-fluorescence studies were performed for the ganglia because many artifacts were imaged due to the autofluorescence of lipofuscin in the neuronal soma. Due to the small number of specimens analyzed, no statistical analysis was carried out.

All experiments were performed in duplicate, and the results in both cases were homogeneous.

## 3. Results

### 3.1. Detection of PIEZO Ion Channels in the Carotid Sinus

Regarding nerves and carotid vessel walls, immunoreactivity for PIEZO1 was observed in the muscle cells of the carotid artery walls and in the nerve profiles and sensory nerve formations of the carotid artery wall. In the carotid sinus adventitia, PIEZO1 was detected exclusively in delicate arborescent nerve endings, free nerve endings, and isolated perivascular nerve fibers but not in the sinus wall. In addition to the nerve profiles and nerve endings, in the muscle layer of the internal and external carotid arteries, PIEZO1-positive isolated nerve fibers and Ruffini-like nerve formations were observed. PIEZO1 was also detected in the nerves of the carotid bifurcation and could sometimes be seen in nerve profiles located in the vascular wall (Figure 1a–d).

Regarding PIEZO2, positive immunoreactivity was observed in the walls of the carotid sinus, where structures remarkably similar to flower-like, arboriform, and Ruffini-like corpuscle endings were formed. In some cases, PIEZO2 immunoreactivity revealed images similar to simple lamellar and Ruffini-like sensory nerve formations. Immunofluorescence for PIEZO2 was also evident in the wall vessels in all layers, but mostly in nerve profiles localized in the common carotid artery division zone (Figure 1e–l). In the bifurcation of the common carotid artery, PIEZO1- and PIEZO2-positive nerve profiles were regularly encountered (Figure 2).

Thus, in the carotid sinus and carotid arteries, PIEZO1 and PIEZO2 are expressed both in nerve profiles, forming different morphotypes of sensory receptors, and in non-nerve cells.

The area occupied by PIEZO1- and PIEZO2-positive nerves amounted to 13.7 ± 3.1% for PIEZO1 and 16.1 ± 1.9% for PIEZO2, suggesting a high density of innervation by nerve profiles immunoreactive for these proteins. The values were similar for all the samples analyzed, without sex- or age-dependent variations.

### 3.2. Detection of PIEZO Ion Channels in the Carotid Body

In the sections of the carotid body that were processed for the simultaneous detection of PIEZO1 and SYN or NSE, which were used as markers for type I glomus cells, most type I glomus cells were unreactive, and only a few cluster cells displayed a very weak but specific immunofluorescence (Figure 3). By contrast, PIEZO1 immunofluorescence was observed in type II glomus cells (Figure 4). In no cases was PIEZO1 immunofluorescence detected in nerve ending profiles inside the carotid body (Figure 5). The area occupied by the PIEZO1-specific immunoreaction amounted to 1.1 ± 0.9% of the total section, and the area occupied by the emergence of both SYN + PIEZO1 and NFP + PIEZO1 amounted to 0.0 and 1.0 ± 0.9% of the total section, respectively (Table 2).

PIEZO2 immunofluorescence in the carotid body was detected in the glomerulus, where a very small subpopulation of type I cells (Figure 6) and most type II glial cells (Figure 6 and Figure 7) were labeled, and in the nerve profiles (Figure 7). The area occupied by the specific immunoreaction of PIEZO2 was 58.1 ± 5.7% of the total section, and the area occupied by the emergence of SYN + PIEZO2 or NFP + PIEZO2 was 1.2 ± 3.9% or 8.4 ± 2.2% of the total section, respectively (Table 2 and Figure 1). The slight discrepancies between the individual values for type I glomus cells and nerves and the sets of type I glomus cells + nerves are likely due to the different antibodies used in immunolabeling.

Therefore, the mechanoproteins PIEZO1 and PIEZO2 are expressed at low levels in the human carotid body. While PIEZO1 was never detected in nerves and only found in a few type I cells, PIEZO2 was detected in nerves and a subpopulation of type I cells. Conversely, both PIEZO1 and PIEZO2 displayed strong immunofluorescence in type II glomus cells. The values were similar for all the samples analyzed, without gender- or age-dependent variations.

### 3.3. Detection of PIEZO Channels in the Petrosal Ganglion and Superior Cervical Sympathetic Ganglion

In humans, the somas of the afferents leading to the carotid sinus and the carotid body reside at the petrosal ganglion of the glossopharyngeal nerve. According to our results, 23.2 ± 8.2% of the neurons were immunoreactive for PIEZO1, and 84.6 ± 7.9% of the neurons were immunoreactive for PIEZO2. Although the sizes of the neurons were not measured, intermediate-sized and large neuroos were the most intensely stained (Figure 8).

On the other hand, both the carotid arterial system and the carotid body received efferent innervation by fibers originating from the superior cervical sympathetic ganglion. In this ganglion, 71.3 ± 7.7% of neurons were immunoreactive for PIEZO1, and almost all (96.1 ± 5.5%) were immunoreactive for PIEZO2 (Figure 8).

Therefore, based on these results, the PIEZO1- and PIEZO2-positive nerve profiles present in the carotid sinus and body, as well as in the walls of carotid arteries, could be both afferent and efferent. However, based on their distribution and the formation of sensory nerve formations, the nerve fibers that were positive for PIEZO1 and PIEZO2 in the carotid sinus should be considered preferentially afferent. As reported above for other parameters, the values were similar for all the samples analyzed, without sex- or age-dependent variations.

## 4. Discussion

The baroreceptor function of the carotid sinus has been well known since the beginning of the last century [30,31,32], although only in the last five years have the molecular mechanisms of mechanotransduction in baroreceptors been elucidated [21,22,33]. On the other hand, although the carotid body is classically regarded as a chemoreceptor [10,34,35], some evidence suggests that it could also participate in some aspects of mechanosensing. This study was conducted to analyze the presence of the mechanoproteins PIEZO1 and PIEZO2, which are multi-pass transmembrane proteins that make up part of the cationic ion channel; they are directly involved in mechanotransduction [4,36,37] in human sinuses and carotid bodies. Based on the expression of PIEZO1 and PIEZO2 in mammalian tissues, it was expected that PIEZO2 would be preferentially localized to nerves and PIEZO1 would be localized to non-nervous tissues. PIEZOs are responsible for converting mechanical cues into biochemical signals in the nervous, cardiovascular, gastrointestinal, respiratory, and urinary systems [38].

The carotid sinus is innervated by myelinated and unmyelinated nerve fibers originating from the carotid sinus nerve, a branch of the glossopharyngeal nerve, whose somas are in the petrosal ganglion. In the carotid sinus, type I (dynamic) carotid baroreceptors have larger myelinated A-fibers; type II (tonic) baroreceptors have smaller A- and unmyelinated C-fibers [27]. Furthermore, the carotid sinus nerve also contains efferent sympathetic fibers [27,39,40]. The afferents in the walls of the carotid sinus, or juxtamural afferents, form different sensory nerve formations that, with rapid and slow adaptation [41], drive information to the central nervous system regarding absolute pressure levels and acute pressure changes [42,43].

Despite decades of research, explanations of baroreceptor mechanosensing and the molecular identity of baroreceptors have remained elusive until recently [33], with Zeng et al. [21] and Min et al. [22] demonstrating the presence of PIEZO1 and PIEZO2 in the aortic and carotid sinuses, respectively. It is known that the functional loss of PIEZO channels causes impaired baroreceptor function and that the PIEZO1 and PIEZO2 channels together, not individually, are critical mechanoreceptors in baroreflex function [21,22,44,45,46,47]. In the present study, we demonstrated that the human carotid sinus and the carotid arteries have terminals of baroreceptors of different morphologies, which are PIEZO1- and PIEZO2-positive. These findings are supported by the presence of PIEZO1 and PIEZO2 in the neurons of the petrosal ganglion, such as in rodents [21,22]. Similarly, the baroreceptors of the aortic sinus located in the nodose ganglia also express PIEZO1 and PIEZO2 [48]. It can, therefore, be said that the PIEZO-positive sensory nerve formations identified in the mural or juxtamural structures of the human carotid sinus are the places where mechanotransduction occurs. They are essential for the transmission of constant information to the central nervous system for the control of heart rate, blood pressure, and respiration [49,50].

Regarding the possible mechanosensory function of the carotid body, based exclusively on the presence of PIEZO proteins in glomus cells, our findings show that only a small population of type I glomus cells express PIEZO2, while they do not express PIEZO1. Furthermore, the glomus nerves also contained PIEZO2 and a very small amount of PIEZO1. Therefore, the putative mechanosensory role of the human carotid body throughout PIEZO channels, if such a role exists, should be played throughout PIEZO2.

A striking finding of our study was the occurrence of PIEZO1 and PIEZO2 in the type II glomus cells. These cells are akin to the satellite cells of the spinal ganglia and the terminal glial cells of sensory corpuscles that express PIEZO1 and PIEZO2 [51,52,53,54,55]. The function of PIEZO proteins in all these glia-like cells remains to be investigated, but it is suggested that they play an active role in the detection of mechanical sensory stimuli. In any case, carotid bodies’ regulation of blood pressure is demonstrated by the fact that the hypotensive response after carotid sinus electrical stimulation is enhanced by the deactivation of carotid chemoreceptors [25], and the short-term chemoreflex and baroreflex regulation of blood pressure are integrated to generate an effector response to a given change in blood pressure [26]. Moreover, hyperactivity of the carotid body has been experimentally shown to trigger hypertension [55,56,57], and the arterial reflex is downregulated when carotid bodies are activated [58]. Conversely, the resection of the carotid body produces sustained reductions in blood pressure [59,60], and its denervation delays the development of hypertension [55,58]. Despite the above experimental data, the role of the carotid body in the regulation of blood pressure should be definitively established in future studies. In addition, it would be of great interest to determine whether this function occurs at the peripheral level (in baroreceptors/mechanoreceptors present in the carotid body) or at the central level (especially in the nucleus of the solitary tract).

The results of this research are limited by the small number of cases available and the fact that we used fixed-paraffin embedded tissues, which prevented us from performing some necessary experiments that would have allowed us to demonstrate that these are indeed mechanosensitive PIEZO channels.

## 5. Conclusions

The present results suggest that PIEZO1 and PIEZO2 are involved in the detection of mechanical stimuli by the carotid sinus, whereas they have a very low level of participation in the detection of mechanical stimuli by carotid bodies, although such a role has not been ruled out, since some populations of type I and type II cells express both proteins. In addition, some nerve endings in the carotid body were positive for PIEZO2 but not for PIEZO1. These findings are compatible with those related to the neurons of the petrosal ganglion. However, since we did not find PIEZO1 in the glomus nerves, it is to be assumed that the PIEZO1-positive neurons of the petrosal ganglion project into territories of the glossopharyngeal nerve other than the carotid body. At present, studies are being conducted in our laboratory to analyze changes in the immunodetection of different potentially mechanosensitive ion channels, especially some isoforms of acid-sensing ion channels and PIEZO channels, in some pathologies in which the carotid sinus and body are involved, such as hypertension or obstructive sleep apnea syndrome.

## Data Availability

The data that support the findings of this study are available from the corresponding author upon reasonable request.

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
