# Peer review of "Immunohistochemical Detection of PIEZO Ion Channels in the Human Carotid Sinus and Carotid Body"

_biomolecules, 2025, doi:10.3390/biom15030386_

Round 1
Reviewer 1 Report
Comments and Suggestions for Authors
Alba et al., submitted this manuscript on potential identification of Peizo channels on human carotid sinus and carotid body. Although their results indicate the presence of Peizo channels, however, there is a significant lack of data to confirm this. This reviewer has some major concerns with this study design and results.
· First, it is worth remembering how to characterize and confirm that it is an ion channel, particularly the ones like Piezo channels that are already well-known and characterized in the literature.
· Based on IHC results, these could be piezo channels, however, these imaging data are not sufficient to claim that.
· Provide Peizo1&2 gene expression as well as protein expression data to confirm the findings.
· To confirm the presence of functional Piezo ion channels, electrophysiological recordings coupled with calcium measurements data are essential.
· Explain why PIEZO1 antibody developed against rat PIEZO1 sequence was used. Show a sequence alignment of PIEZOS across human, mice and rats to clarify how their sequences match, especially the epitope/target for the antibody used.
· Clarify how quantitative analysis was done on five sections per sample/subject. Did the study perform analysis for all 8 subjects, or just 5 sections from 1 subject?
· Provide exact number of experimental replicates and biological replicates in each figure legends.
· Provide indication of what each panel represents in all figures and figure legends.
· Imaging data should be quantified and plotted for comparative analysis.
· A sex difference analysis could be performed.
Comments on the Quality of English Language
Spelling error
Author Response
The authors thank the anonymous referee for his/her comments that will undoubtedly contribute to improving the quality of our study.
Alba et al. submitted this manuscript on potential identification of Piezo channels on human carotid sinus and carotid body. Although their results indicate the presence of Piezo channels, there is a significant lack of data to confirm this. This reviewer has some major concerns with this study design and results.
First, it is worth remembering how to characterize and confirm that it is an ion channel, particularly the ones like Piezo channels that are already well-known and characterized in literature.
Thank you for your consideration. Indeed, we agree with you that the title could be misleading and lead to the belief that functional studies had been carried out to demonstrate the presence of functional PIEZO channels in the carotid breast and body. That is why we have changed the title to indicate that it is an immunohistochemistry study on human tissues.
Based on IHC results, these could be piezo channels, however, these imaging data are not sufficient to claim that.
In line with the response to his general comment, throughout the manuscript there is talk of immunoreaction for these channels.
Provide Piezo1&2 gene expression as well as protein expression data to confirm the findings.
Although the experiments he proposes are important, with the material available at the moment it is not possible to carry them out.
To confirm the presence of functional Piezo ion channels, electrophysiological recordings coupled with calcium measurements data are essential.
I totally agree, but this is not the aim of our work. It is not possible to carry out this type of study on fixed biobank material.
Explain why PIEZO1 antibody developed against rat PIEZO1 sequence was used. Show a sequence alignment of PIEZOS across human, mice and rats to clarify how their sequences match, especially the epitope/target for the antibody used.
In this case, there was an error in the wording. In our laboratory we have several antibodies against PIEZO1 that react in different species. The antibody used was against a human PIEZO1 sequence, and not a rat one. The error has been corrected in the revised manuscript.
Clarify how quantitative analysis was done on five sections per sample/subject. Did the study perform analysis for all 8 subjects, or just 5 sections from 1 subject?
The section on the Quantitative Analysis of Material and Methods has been almost completely rewritten. The quantitative analysis was carried out in 5 sections for each of the 8 subjects, and in each section, 10 fields were quantified; that is: 50 fields per subject x 8 subjects. In the new version of the manuscript it is described in detail.
Provide exact number of experimental replicates and biological replicates in each figure legends.
In our laboratory, the norm is to always do immunohistochemistry experiments twice.
Provide indication of what each panel represents in all figures and figure legends.
We believe that the information provided in the figure legend is sufficient to supplement what is described in the Results. The journal's Author Services has reviewed and edited the manuscript and figures and deemed them appropriate.
Imaging data should be quantified and plotted for comparative analysis.
As previously stated, the quantitative analysis section has been drafted ex novo.
Reviewer 2 Report
Comments and Suggestions for Authors
The authors investigate the expression of the PIEZO 1 and 2 ion channel proteins by means of immunohistochemistry. They followed up the assertion that PIEZO channels are expressed in the carotid bodies and that they mediate a baroreflex there.
They found that PIEZO 1 and 2 are less expressed in the glomus cells and the corresponding nerve endings than in the carotid sinus vessels. This is interesting regarding knowledge about the function of the carotid bodies.
Nevertheless, the conclusion, written in the abstract is difficult to understand. In detail, the data given in lines 37 – 40 do not directly support the following conclusion. The conclusions are more comprehensible in the “Conclusions” chapter and should be changed accordingly in the abstract. The conclusion that “the putative role as a mechanosensor/mechanotransducer of the carotid body, based on the occurrence of PIEZO ion channels, should be almost null.” should be drawn with caution.
Further points:
- Some grammatical mistakes should be corrected, for example, L 33, 34, 269.
- Quantitative data from the sinus caroticus preparations are needed, even if the number of measurements are too small for statistical evaluation. The statements
“This study was designed to analyze the presence of the mechanoproteins PIEZO1 and PIEZO, which are multipass transmembrane proteins forming a part of cationic ion channels, directly involved in mechanotransduction in the human sinus and carotid body.” (L277) and
“A quantitative image analysis was carried out in the carotid body, petrosal ganglia, and superior cervical sympathetic ganglia…” (L156) are contradictory in this regard.
- Please, explain in more detail the composition of saline solution and PBS.
Comments on the Quality of English Languagesee above
Author Response
The authors thank the anonymous referee for his/her comments that will undoubtedly contribute to improving the quality of our study.
The authors investigate the expression of the PIEZO 1 and 2 ion channel proteins by means of immunohistochemistry. They followed up the assertion that PIEZO channels are expressed in the carotid bodies and that they mediate a baroreflex there.
They found that PIEZO 1 and 2 are less expressed in the glomus cells and the corresponding nerve endings than in the carotid sinus vessels. This is interesting regarding knowledge about the function of the carotid bodies.
Nevertheless, the conclusion written in the abstract is difficult to understand. In detail, the data given in lines 37 – 40 do not directly support the following conclusion. The conclusions are more comprehensible in the “Conclusions” chapter and should be changed accordingly in the abstract. The conclusion that “the putative role as a mechanosensor/mechanotransducer of the carotid body, based on the occurrence of PIEZO ion channels, should be almost null.” should be drawn with caution.
Thank you very much for your comment. The final part of the Abstract has been modified to meet the consideration of the reviewer.
Further points:
Some grammatical mistakes should be corrected, for example, L 33, 34, 269.
In the revision, the manuscript has been reviewed and edited by the journal's Author Services to correct all these errors.
Quantitative data from the sinus caroticus preparations are needed, even if the number of measurements are too small for statistical evaluation.
The statements “This study was designed to analyze the presence of the mechanoproteins PIEZO1 and PIEZO, which are multipass transmembrane proteins forming a part of cationic ion channels, directly involved in mechanotransduction in the human sinus and carotid body.” (L277) and “A quantitative image analysis was carried out in the carotid body, petrosal ganglia, and superior cervical sympathetic ganglia…” (L156) are contradictory in this regard.
Thanks for the suggestion. The suggested study has been carried out and added to the Results.
Please, explain in more detail the composition of saline solution and PBS.
All the manufacturer details were added to the revised version of the manuscript.
Round 2
Reviewer 1 Report
Comments and Suggestions for Authors
Based on authors’ responses, it is understandable that fresh tissue is not available for other experiments. However, authors are encouraged to collect the snap-frozen materials, if possible/available, and perform western blotting experiments, because confirmatory western blot data would make the study stronger.
Authors responded, “In our laboratory, the norm is to always do immunohistochemistry experiments twice.” Authors are encouraged to mention that in the method section.
Authors responded, “As previously stated, the quantitative analysis section has been drafted ex novo.” Authors are highly suggested to plot the imaging data (intensity/percentage) and make graphs, which will make the manuscript easier to understand for the readers.
The limitations of the study at the end of discussion is a good addition.
No response is provided to the last comment about sex difference.
Author Response
Based on authors’ responses, it is understandable that fresh tissue is not available for other experiments. However, authors are encouraged to collect the snap-frozen materials, if possible/available, and perform western blotting experiments, because confirmatory western blot data would make the study stronger.
We thank the anonymous reviewer for their understanding and advice. We have tried to get fresh and/or frozen material but, at moment, we have not been able to do so. The material we have in the Biobank comes mostly from what is obtained during organ removal for transplantation. We are waiting to obtain carotid bodies in ideal conditions to carry out molecular biology studies and ensure the results obtained by immunohistochemistry not only in PIEZOs, but also in other ion channels. We are currently developing a project on the multi-sensory function of the carotid body and the experiments you suggest are mandatory to reach solid conclusions; but no, the moment is not possible.
Authors responded, “In our laboratory, the norm is to always do immunohistochemistry experiments twice.” Authors are encouraged to mention that in the method section.
Thanks for the advice; it has been done.
Authors responded, “As previously stated, the quantitative analysis section has been drafted ex novo.” Authors are highly suggested to plot the imaging data (intensity/percentage) and make graphs, which will make the manuscript easier to understand for the readers.
Thank you very much for the suggestion. A graph has been added to accompany Table 1. In both cases, only the area occupied by the immunoreaction is accompanied. Immunolabeling intensity has not been assessed because, as you are well aware of in immunohistochemistry, unlike other methods, there is no stoichiometric relationship between the amount of antigen and the intensity of the immunoreaction.
The limitations of the study at the end of discussion is a good addition.
Thanks a lot.
No response is provided to the last comment about sex difference.
A brief sentence has been included at the end of the results to express this detail.
Reviewer 2 Report
Comments and Suggestions for Authors
On page 10, the authors state that “only a small population of type I glomus cells express PIEZO2” whereas in Table 2 the information is given that the percentage of area occupied by PIEZO2 in Type I glomus cells is 50 %. This is a contradiction.
In this regard, the statement in the abstract “The results of this study suggest that PIEZO1 and PIEZO2 could be involved in the detection and/or mechanotransduction of the human carotid sinus, whereas the putative role of the carotid body as a primary mechanosen-sor/mechanotransducer, based exclusively on the occurrence of PIEZO ion channels, should be almost null …” is not justified since PIEZO2 channels might be involved in mechanotransduction due to their expression in glomus Type I cells.
The authors need to clarify this.
Author Response
On page 10, the authors state that “only a small population of type I glomus cells express PIEZO2” whereas in Table 2 the information is given that the percentage of area occupied by PIEZO2 in Type I glomus cells is 50 %. This is a contradiction.
The authors would like to thank the reviewer for the great detail with which he/her has reviewed the manuscript. There is indeed a serious error and inconsistency between the wording and the table, due to the wording in R1 of the methodology of the measurements and the non-modification in the table. It has now been carefully reviewed and corrected; in addition, a graph has been added that accompanies Table 1 to facilitate understanding.
The statement at the end of the Abstract was modified.